The profound impact of COVID-19 on college students’ physical fitness

http://orcid.org/0000-0002-1720-9850 Sun Jianzhong 1 2
http://orcid.org/0000-0002-1720-9850 Qiao Bin 3
Lin Chan 1 2 256545797@qq.com
1 School of Physical Education, Chizhou University , Chizhou , China
2 Sports for Health Promotion Center, Chizhou University , Chizhou , China
3 Basci Teaching Department, Chizhou Vocational and Technical College , Chizhou , China
Anson Lesley
Electronic publication date: 2025 Oct 23
Publication date: 2025
Volume: 13
Electronic Location ID: e20293
Received 2025 Jun 10; Accepted 2025 Oct 3
Copyright: © 2025 Sun et al.
Copyright year: 2025
Copyright holder: Sun et al.
License: This is an open access article distributed under the terms of the Creative Commons Attribution License, which permits unrestricted use, distribution, reproduction and adaptation in any medium and for any purpose provided that it is properly attributed. For attribution, the original author(s), title, publication source (PeerJ) and either DOI or URL of the article must be cited.
License URL: https://creativecommons.org/licenses/by/4.0/

Keywords: Medical college, COVID-19, Physical fitness, Profound influence, Post-epidemic

Funding: Chizhou University High-Level Talent Research CZ2022YJRC03 Chizhou University Scientific Research Institutions KYJG012 Ministry of Education Humanities and Social Sciences Research Youth Project 23YJC890002 This work was supported by the Chizhou University High-Level Talent Research Startup Fund(CZ2022YJRC03), Chizhou University scientific research institutions (Grant No. KYJG012), and the Ministry of Education Humanities and Social Sciences Research Youth Project Approval of 2023 (Award No. 23YJC890002). The funders had no role in study design, data collection and analysis, decision to publish, or preparation of the manuscript.

==============================
Background

This study expands existing research by examining longitudinal impacts of the COVID-19 pandemic on medical college students’ physical fitness.

Methods

A medical college in Wenzhou, was selected to examine the changes in physical fitness indicators among students from 2019 to 2021.

Results

While most students maintained normal weight status (85.2%), overweight/obese prevalence increased significantly (8.0% in 2019, 8.9% in 2020, and 10.1% in 2021). Among male students, 67.0% were classified within the passing range, while the majority of female students (55.0%) fell within the “good” grade category. In 2021, a significant decline was observed in the standing long jump, 50-m dash, and 1,000/800 m run (p < 0.05) across both genders.

Conclusions

The lingering effects of the COVID-19 pandemic have significantly contributed to increased weight gain among college students and a decline in their endurance running performance.

Introduction

The COVID-19 pandemic precipitated global societal disruption (Podstawski et al., 2022; Drenowatz et al., 2023; Sun et al., 2023), compelling nationwide implementation of containment policies (Güner, Hasanoğlu & Aktaş, 2020; Ingram et al., 2021). In China, stringent containment strategies were enforced (Cheng et al., 2023; Kraemer et al., 2020), which included the categorization of regions into containment zones, control areas, and prevention areas. From December 2019 to December 2022, the epidemic spread unpredictably across the country (Zhu, Wei & Niu, 2020; Zheng, Liu & Lu, 2023), with strict monitoring of inter-provincial travel significantly disrupting the daily lives and work of people nationwide. Colleges and universities, as densely populated institutions, implemented a range of measures during the pandemic, including campus closures (Xu et al., 2023), online teaching, daily health monitoring, and regular nucleic acid testing (Peng et al., 2020). Some institutions also experienced severe outbreaks of COVID-19, which had a profound impact on students’ academic performance, personal lives, and psychological well-being (Peng et al., 2024).

Since January 8, 2023, China’s epidemic situation has been largely resolved (Chirico & Teixeira da Silva, 2023); however, its effects continue to persist (Sarker et al., 2023). The pandemic has fostered international collaboration and led to significant shifts in lifestyle practices, including the rapid expansion of online shopping and e-commerce platforms, as well as the growing prevalence of online meetings. Additionally, it has become increasingly common for college students to rely on take-out food (Jang, Lee & Jung, 2022), spend extended periods using electronic devices (Schmidt et al., 2020), and adopt more sedentary lifestyles (Rahman et al., 2020; Dziewior et al., 2024), thereby reducing interpersonal interactions and negatively impacting their physical fitness levels. These changes in daily life and work patterns require additional time for recovery and a renewed emphasis on physical activity to ultimately improve overall physical fitness (Zeng et al., 2023; Thomas et al., 2023).

Most scholars have focused on the changes in physical fitness before and after the epidemic, particularly those observed in 2019 and 2020 (Xia et al., 2021; Kowalsky et al., 2023), yet critical knowledge gaps persist regarding delayed physiological sequelae. Few studies have examined physical parameters and indicators from 2021 to 2024 (Sun et al., 2023). Additionally, the physical fitness indicators used in existing research are often limited, making it difficult to fully capture the comprehensive effects of the pandemic on physical health (Feng et al., 2023).

This study employs a more comprehensive set of nine physical fitness indicators to investigate the broad impact of the COVID-19 pandemic on college students from 2019 to 2021. The research has two main objectives: first, to gather specific data on various physical indices over the past 3 years (2019–2021) and identify significant differences influenced by the pandemic; second, to analyze the trends of these indicators following the outbreak and explore the long-term effects and lag impact of the COVID-19 epidemic. Corresponding, this study examined three research hypotheses: (1) the COVID-19 pandemic variably affected all dimensions of college students’ physical fitness before and after its onset; (2) reduced physical exercise contributed to increased obesity rates; (3) the pandemic exerted profound and persistent adverse effects on students’ health.

Materials and Methods

Participants and data source

The study selected all students from a specialized medical college in Wenzhou City as participants. The number of students involved in the survey from 2019 to 2021 was 11,307, 11,562, and 12,282, respectively, for a total of 35,151 students. Participants were aged between 17 and 23 years. Eight physical fitness test indicators were used, including height, weight, vital capacity, standing long jump, sit-and-reach, 50-m dash, pull-ups (for males), sit-ups (for females), and 1,000 m (for males) and 800 m (for females) runs. All students provided written informed consent, with minors obtained written informed consent from their guardians. This study was approved by the Human Experimental Ethics Committee of Chizhou University (No. CZ2018YJRC01).

Physical exercise mode of students during the epidemic period

During the pandemic, academic continuity was maintained through virtual instruction despite campus closures. Professional courses and medical humanities education were delivered simultaneously through online teaching methods. Instructors adopted a hybrid approach to maintain the quality of online instruction while incorporating ideological and political components into the curriculum.

For on-campus activities, the Wisdom Tree Network course platform was used to deliver physical fitness courses to college students. The process involved teachers recording instructional videos for class content (e.g., Ba Duan Jin), which were then converted into short videos and shared in class groups. Students uploaded their own exercise videos, which teachers reviewed and annotated. During holidays and breaks, the strategy of “classes suspended but teaching continues, stay home but learning persists” was implemented. This required students to engage in online “Internet Plus Home Fitness” activities. The sessions consisted of a warm-up phase (primarily dynamic stretching), followed by the main exercises (such as plank holds, side leg lifts, dry land breaststroke, squat jumps, mountain climbers, burpees, etc.), and concluding with relaxation (mainly static stretching).

Measurement of physical fitness indicators

All physical fitness indicators in this study were carefully assessed in accordance with the National Student Physical Health Standard (The latest edition in 2014). To ensure the accuracy of the test data, a standardized measuring instrument was used, which was calibrated before each measurement and corrected three times daily. All physical fitness examiners, including both teachers and students, underwent rigorous pre-training. The annual physical fitness test was conducted each October, with testing sessions held from 7:30 to 11:30 am and from 2:30 to 5:30 pm daily. The measurement techniques for the various physical fitness indicators were consistent with those outlined in my previous publication in PLOS ONE (https://doi.org/10.1371/journal.pone.0308483) (Sun et al., 2024).

Data statistical analysis

Incomplete or invalid physical measurement data were excluded from the analysis. The physical fitness indicator test data were statistically analyzed using SPSS 25.0 (SPSS, Inc., Chicago, Illinois, USA). First, the normality of the physical fitness indicators was assessed using the Kolmogorov-Smirnov test. Descriptive statistics were calculated for age distribution, body mass index (BMI) categories, and the total score of the physical examination. Quantitative data were described using appropriate measures. Normality testing via Kolmogorov-Smirnov tests indicated non-normal distribution for all physical fitness metrics across three consecutive years (p < 0.001). Accordingly, the Kruskal-Wallis H test was applied to compare the differences in physical fitness indicators over 3 years.

Results

Participants characteristics

In 2019, there were 11,307 students (4,004 male and 7.303 female), in 2020, 11,562 students (4,198 male and 7,364 female), and in 2021, 12,282 students (4,629 male and 7,653 female). In total, 35,151 students (12,831 male and 22,320 female) provided valid physical fitness test data for this study (Table 1). The participants were aged 17 to 23, with the majority falling between 18 and 21 years old. According to BMI classifications, most students had normal weight, with 75.8% to 78.5% of male students and 88.1% to 90.9% of female students falling into this category. Overweight students represented the second-largest proportion, comprising 11.7% to 13.2% of male students and 3.8% to 4.9% of female students. Underweight students ranked third, accounting for 6.2% to 7.4% of males and 3.8% to 6.4% of females. Obese students showed the lowest prevalence (males: 2.7–3.8%; female: 0.7–1.1%).

Table 1 Classification evaluation of physical fitness tests from 12,831 male and 22,320 female records.

Variables	Total	Male	Female	
		2019	2020	2021	2019	2020	2021	2019	2020	2021	
Age (years)
[n (%)]	≤17	89 (0.7)	92 (0.7)	79 (0.6)	25 (0.6)	30 (0.7)	27 (0.5)	64 (0.8)	62 (0.8)	52 (0.6)	
18	2,000 (17.6)	2,049 (17.7)	2,067 (16.8)	733 (18.3)	744 (17.7)	823 (17.7)	1,267 (17.3)	1,306 (17.7)	1,244 (16.2)	
19	3,046 (26.9)	2,921 (25.2)	3,231 (26.3)	1,012 (25.2)	1,113 (26.5)	1,250 (27.0)	2,034 (27.8)	1,808 (24.5)	1,981 (25.8)	
20	2,748 (24.3)	2,944 (25.4)	2,975 (24.2)	959 (23.9)	1,012 (24.1)	1,142 (24.6)	1,789 (24.4)	1,932 (26.2)	1,833 (23.9)	
21	2,369 (20.9)	2,524 (21.8)	2,816 (22.9)	856 (21.3)	911 (21.7)	963 (20.8)	1,513 (20.7)	1,613 (21.9)	1,853 (24.2)	
22	911 (8.0)	946 (8.1)	978 (7.9)	355 (8.8)	352 (8.3)	365 (7.8)	566 (7.6)	594 (8.0)	613 (8.0)	
≥23	144 (1.2)	86 (0.7)	136 (1.1)	64 (1.5)	36 (0.8)	59 (1.2)	80 (1.0)	50 (0.6)	77 (1.0)	
Total participant	11,307	11,562	12,282	4,004	4,198	4,629	7,303	7,364	7,653	
BMI Status
[n (%)]	Underweight	630 (5.5)	742 (6.4)	622 (5.0)	299 (7.4)	264 (6.2)	327 (7.0)	331 (4.5)	478 (6.4)	295 (3.8)	
Normal weight	9,764 (86.3)	9,786 (84.6)	10,402 (84.6)	3,123 (77.9)	3,296 (78.5)	3,513 (75.8)	6,641 (90.9)	6,490 (88.1)	6,889 (90.0)	
Overweight	750 (6.6)	814 (7.0)	991 (8.0)	471 (11.7)	501 (11.9)	613 (13.2)	279 (3.8)	313 (4.2)	378 (4.9)	
Obesity	163 (1.4)	220 (1.9)	267 (2.1)	111 (2.7)	137 (3.2)	176 (3.8)	52 (0.7)	83 (1.1)	91 (1.1)	
Total Score
[n (%)]	Excellent	722 (6.3)	726 (6.2)	698 (5.6)	215 (5.3)	223 (5.3)	193 (4.1)	507 (6.9)	503 (6.8)	505 (6.5)	
Good	4,863 (43.0)	5,238 (45.3)	4,876 (39.7)	857 (21.4)	950 (22.6)	903 (19.5)	4,006 (54.8)	4,288 (58.2)	3,973 (51.9)	
Pass	5,390 (47.6)	5,286 (45.7)	6,146 (50.0)	2,689 (67.1)	2,779 (66.1)	3,139 (67.8)	2,701 (36.9)	2,507 (34.0)	3,007 (39.2)	
Fail	322 (2.9)	312 (2.6)	562 (4.5)	243 (6.0)	246 (5.8)	394 (8.5)	89 (1.2)	66 (0.8)	168 (2.1)	

The overall pass rate for the physical examination ranged from 45.7% to 50.0%, with 39.7% to 45.3% of students achieving a “good” rating, 5.6% to 6.3% an “excellent” rating, and 2.6% to 4.5% failing. For male students, the pass rate was higher, ranging from 66.1% to 67.8%, followed by 19.5% to 22.6% categorized as “good,” 5.8% to 8.5% failing, and 4.1% to 5.3% achieving an “excellent” rating. For female students, the distribution was as follows: 51.9% to 58.2% rated “good,” 34.0% to 39.2% passing, 6.5% to 6.9% rated “excellent”, and 0.8% to 2.1% failing.

Score and comparison of physical fitness indicators

For both male and female students, nine physical fitness test parameters were assessed, with specific scores presented in Table 2. As shown in Figs. 1 and 2, the average values and 95% confidence intervals (CIs) for these nine physical fitness indicators were analyzed over the years 2019 to 2021. Integrated analysis of longitudinal data (Table 2, Figs. 1, 2) indicates progressive annual increase in height, weight, BMI, and sit-and-reach performance. For male students, the standing long jump and 50-m dash demonstrated a downward trend over time, while vital capacity, pull-ups, and the 1,000 m running demonstrated transient improvement followed by progressive deterioration. In contrast, for female students, the standing long jump and 800 m run exhibited a downward trend, while vital capacity, the 50-m dash, and sit-ups showed an initial increase and followed by a decrease.

Table 2 Comparison of mean (±SD) and difference of physical fitness test indexes by gender.

Gender	Items	2019 (A)	2020 (B)	2021 (C)	A–B	A–C	B–C	
Male	Height (cm)	174.63 ± 5.58	174.75 ± 5.65	175.17 ± 5.59	0.55	0*	0.006*	
Weight (kg)	64.84 ± 10.09	65.52 ± 10.11	65.99 ± 10.66	0.006*	0*	0.27	
BMI (kg/m2)	21.23 ± 2.92	21.42 ± 2.92	21.48 ± 3.10	0.017*	0.002*	1.00	
Vital capacity (ml)	4,825.16 ± 710.44	4,851.77 ± 688.05	4,816.10 ± 709.53	0.14	
Standing long jump (cm)	228.84 ± 22.74	228.11 ± 22.48	227.15 ± 23.29	0.31	0*	0.08*	
Sit-and-reach (cm)	13.47 ± 7.15	13.68 ± 7.07	13.84 ± 7.22	>0.05	
50-m dash (s)	7.27 ± 0.51	7.29 ± 0.62	7.35 ± 0.58	0.72	0*	0*	
Pull-ups (count)	6.82 ± 6.46	6.89 ± 6.47	6.26 ± 6.22	1.0	0*	0*	
1,000 m (s)	240.68 ± 30.99	238.88 ± 32.99	247.36 ± 36.51	0*	0*	0*	
Female	Height (cm)	161.97 ± 5.27	162.20 ± 5.31	162.45 ± 5.33	0.02*	0*	0.013*	
Weight (kg)	52.40 ± 6.60	53.08 ± 6.87	53.30 ± 7.11	0*	0*	0.23	
BMI (kg/m2)	19.95 ± 2.14	20.16 ± 2.28	21.18 ± 2.36	0*	0*	1.0	
Vital capacity (ml)	3,336.72 ± 500.55	3,377.31 ± 469.86	3,363.63 ± 499.42	0*	0.001*	1.0	
Standing long jump (cm)	173.47 ± 16.90	172.70 ± 17.26	171.79 ± 18.19	0.099	0*	0.008*	
Sit-and-reach (cm)	18.81 ± 6.19	18.96 ± 6.04	19.49 ± 6.23	0.59	0*	0*	
50-m dash (s)	8.91 ± 0.68	8.90 ± 0.65	8.95 ± 0.70	1.0	0.008*	0.005*	
Sit-ups (count)	43.25 ± 9.59	43.37 ± 9.53	43.02 ± 9.78		0.061		
800 m (s)	208.76 ± 18.22	225.19 ± 26.71	233.60 ± 31.50	0*	0*	0*	
Note:

A–B, A–C, B–C mean the p-values of Kruskal-Wallis H test between 2019–2020, 2019–2021, and 2020–2021.

Figure 1 The mean and 95% CI of items of the physical fitness indicators on male from 2019 to 2021.

Figure 2 The mean and 95% CI of items of the physical fitness indicators on female from 2019 to 2021.

Using non-parametric tests for pairwise comparisons, significant differences in several physical fitness indicators between 2019 and 2020 were identified. For males, these differences were observed in weight, BMI, and the 1,000 m run. For females, significant changes were found in height, weight, BMI, vital capacity, and the 800 m run. In contrast, between 2019 and 2021, significant changes were observed in all nine indicators for males, except for vital capacity and sit-and-reach. For females, significant changes were found in all nine indicators, except for sit-ups. Furthermore, comparisons between 2020 and 2021 revealed significant differences in the following physical fitness indicators: for males, height, standing long jump, 50-m dash, pull-ups, and the 1,000 m run; and for females, height, standing long jump, sit-and-reach, 50-m dash, and the 800 m run.

Discussion

To test Hypothesis 1, longitudinal changes in collegiate BMI measurements pre- to post-pandemic were analyzed, utilizing standardized national assessment protocols. BMI categories among college students in Wuhan during the period 2019–2020 were as follows: normal weight, 58.9%; underweight, 17.5%; overweight, 15.1%; and obesity, 8.5% (Xia et al., 2021). Our study reveals a relatively higher proportion of normal-weight individuals and a lower prevalence of overweight and obesity, suggesting that college students in Wenzhou were less affected by the COVID-19 epidemic compared to those in Wuhan. This observation can be attributed to Wuhan’s role as the epicenter of the epidemic in China. The pandemic, which was particularly severe in December 2019, did not subside until April 2020, leading to a more pronounced impact on the physical health of Wuhan’s college students. As a result, there were fewer normal-weight individuals and a higher prevalence of overweight and obesity in that region. By comparison, Anhui college students exhibited distinct BMI distribution: underweight (15.6%), normal weight (62.6%), overweight (15.6%), and obese (6.3%) (Sun et al., 2023). The prevalence of overweight and obesity in this study was notably lower than in Anhui. This discrepancy may be attributable to several factors, including the involvement of different research groups and regional variations. Our study focused on Zhejiang Province, a coastal area with distinct dietary habits, such as a higher consumption of seafood. This could contribute to a leaner physical condition, which may have made students in coastal cities more resilient to the extensive impacts of the epidemic. Additionally, data from the China Obesity Map show that the obesity rate in coastal regions is lower compared to inland areas (Zhejiang province 4.9% vs. Anhui province 9.1%) (Yang et al., 2017).

Hypothesis 2 was tested through analysis of pandemic impacts on collegiate physical fitness with specific focus on weight-related metrics, including overweight/obesity prevalence and BMI trajectories. From 2019 to 2021, our study observed an increasing trend in the proportion of overweight and obese students. This finding aligns with the results of Jiang et al. (2024) who reported a similar year-by-year rise in the prevalence of overweight and obesity among male students in Hunan (from 18.08% in 2019 to 19.12% in 2020 to 19.54% in 2021). These trends demonstrate that the impact of the COVID-19 epidemic on college students’ weight was not temporary but has had lasting effects. In our study, the rates of overweight and obesity were 15.5% for male students and 5.1% for female students, with the proportion of overweight and obese males being more than three times higher than that of females. This finding was consistent with data from southern Anhui, where the rates of overweight and obesity were 28.0% for males and 12.7% for females (Sun et al., 2023), as well as from Wuhan, where the respective rates were 16.2% for males and 7.4% for females (Xia et al., 2021). These results underscore the significant impact of the epidemic on students’ weights, particularly its exacerbating effect on overweight and obesity among male students.

To substantiate hypotheses 1 and 2, stratified analyses of pandemic-induced physical fitness changes across four performancetiers were conducted: failing, passing, good, and excellent (based on national test scoring standards). The failure rate for physical examinations in this study decreased from 2.9% in 2019 to 2.6% in 2020, but then increased to 4.5% in 2021. Similarly, Jiang et al. (2024) reported a rising failure rate in Hunan Province, with the “fail” percentage increasing from 8.99% in 2019 to 10.33% in 2020 and reaching 12.94% in 2021. This suggests that the impact of the epidemic on physical fitness continues to be felt, with the highest failure rates occurring in 2021. These findings underscore the necessity for longitudinal investigations (2022–2024), to quantify persistent epidemic sequelae on physical fitness. Our data indicate most students passed the physical examination, with male students primarily scoring within the pass range (66.1–67.8%) and female students scoring in the “good” range (51.9–58.2%). Jiang et al. (2024) also found that physical examination scores for college students in Hunan Province over the 3 years were predominantly in the passing range (74.1–81.2% for males; 74.01–79.7% for females), which is consistent with the results of this study. Those findings suggest that, relative to females, males exhibited poorer physical fitness, likely due to their greater involvement in outdoor activities, which were significantly impacted by epidemic control measures (Dziewior et al., 2024; Feng et al., 2023). Conversely, females typically engaged in less physical activity and were less affected by these measures (Jiang et al., 2024; Zhao et al., 2024; Hu et al., 2022).

Hypotheses 2 and 3 were substantiated through longitudinal analysis of pandemic-associated weight trajectories among college students, with particular focus on clinically significant BMI elevations and their enduring health implications. It was observed that the prevalence of overweight among male students (12.3%) was higher than that among female students (4.3%). This finding aligns with studies by Sun et al. (2023) (overweight prevalence of males and females: 19.9% vs. 9.2%) and Xia et al. (2021) (overweight prevalence of males and females: 19.5% vs. 10.8%). Similarly, Jiang et al. (2024) reported a significantly higher overweight rate in male students (17.0–20.7%) compared to female students (8.3–10.1%) in Hunan. Additionally, our study identified significant gender disparity in obesity prevalence (males: 3.2% vs. females: 1.0%), which was consistent with the findings of Sun et al. (2023) (obesity rate for males and females: 8.2% vs. 3.5%) and (Xia et al., 2021) (obesity rate for males and females: 12.9% vs. 4.1%). Li et al. (2022) also observed that the ratio of overweight to obesity was significantly higher in male students than in female students, which is in line with our conclusions. Jiang et al. (2024) noted a much higher obesity rate in male students (6.8–8.1%) compared to female students (2.53–2.96%) in Hunan. These data suggest that males have been more severely affected by the COVID-19 epidemic, as the restrictions imposed to control the virus significantly reduced outdoor exercise opportunities for males. This has had a profound impact on their health, particularly in relation to obesity.

Hypothesis 2 was substantiated: Student overweight/obesity prevalence demonstrated significant associations with reduced physical activity and increased sedentary behavior during the pandemic. Overweight and obesity were closely associated with the sharp decline in physical activity levels and the corresponding increase in sedentary behavior during the COVID-19 epidemic (Dziewior et al., 2024; Yu et al., 2022). Studies have shown that physical activity levels were significantly reduced during this period. For example, Rahman et al. (2020) observed that physical activity among young adults decreased by 42.5%. Similarly, Ammar et al. (2020) reported a 28% increase in sedentary time during the epidemic. Pandemic-induced weight dysregulation was significantly associated with altered dietary patterns. Multiple studies indicated a surge in the consumption of fast foods and processed foods, coupled with a decline in the intake of fruits and vegetables, which contributed to the rise in overweight and obesity rates (Esobi, Lasode & Barriguete, 2020). Concurrently, pandemic-related psychosocial stressors (panic, anxiety) may led to emotional eating, driving non-hunger-related consumption (Wilson et al., 2021), and subsequent weight accretion (Nutley et al., 2021). Han et al. (2023) found that physical activity levels were significantly negatively correlated with depression and anxiety (p < 0.01), highlighting the psychological factors that may have contributed to changes in weight during the pandemic.

Hypothesis 1 was substantiated: The pandemic induced differential deterioration across all physical fitness domains among college students. There was little change in performance for the standing long jump and 50-m dash (Sun et al., 2023; Xia et al., 2021), as these events involve short bursts of explosive activity and are less dependent on outdoor exercise. Consequently, they were less affected by the COVID-19 epidemic (Jiang, Wu & Zhang, 2023; Qin & Wen, 2023). Additionally, male students’ performance in vital capacity, pull-ups, and the 1,000 m run showed an initial increase followed by a decline. For example, the 1,000 m run time improved by 1.8 s in 2020 but then worsened by 8.8 s in 2021, while vital capacity first increased by 27.0 ml in 2020 and then decreased by 36.0 ml in 2021. This pattern suggests a significant positive association between 1,000 m running and vital capacity. Xia et al. (2021) found that the 1,000 m time for male students decreased by 14 s in 2020, which contrasts with the findings of this study. This discrepancy likely reflects Wuhan’s status as the pandemic epicenter, where endurance events were significantly impacted by the lockdown from December 2019 to April 2020. Supporting this contextual interpretation, Feng et al. (2022) documented significant performance deterioration at Tsinghua University, male students’ performance in the 1,000 m run and pull-ups declined by 10.91% (p < 0.001) and 23.89% (p < 0.001), respectively, in 2020 compared to 2019. These findings further underscore that endurance running, were among the most severely affected by the COVID-19 epidemic (Sun et al., 2023; Zhang, Liu & Li, 2023; Cao, 2023). Without access to outdoor exercise, the performance of male students in the 1,000 m run sharply declined, reflecting the broader impact of the pandemic on physical fitness.

Consistent with Hypotheses 1 and 3, female college students demonstrated significantly compromised endurance running performance attributable to pandemic-related disruptions. In this study, it was observed that the standing long jump and 800 m run performance among female students exhibited a downward trend from 2019 to 2021, with the 800 m run times increasing by 16.5 s in 2020 and 8.4 s in 2021. Additionally, vital capacity, 50-m dash, and sit-up performance initially improved but later declined. Xia et al. (2021) documented an 11-s deterioration in female students 800 m run performance in Wuhan during 2020, aligning with the trends observed in this study. In contrast, Sun et al. (2023) found that the standing long jump, 50-m dash, and sit-ups all showed an increasing trend over the years, while the 800 m run decreased initially and then increased, which was inconsistent with the conclusions of this study. This discrepancy divergence likely stems from distinct sampling approaches: both Xia et al. (2021) and this study used data from the entire student population, while Sun et al. (2023) followed the same sample of students over three consecutive years. Furthermore, Feng et al. (2023) reported significant decreases in the 800 m run (7.97%) and sit-ups (4.91%) among female students at Tsinghua University in 2020, indicating that the pandemic had a substantial impact on female students’ endurance running events as well.

Conclusion

This study accomplished its dual objectives by: (1) substantiating three research hypotheses concerning differential pandemic impacts on university students’ physical fitness; (2) documenting persistent adverse effects with significant temporal disruption; and (3) delineating critical future research trajectories. From 2019 to 2021, medical college students in Zhejiang province experienced a year-on-year increase in BMI, sit-and-reach performance, and the proportion of students failing the physical fitness test. The prevalence of overweight and obesity also rose significantly, and performance in endurance events, such as the 800 and 1,000 m runs, deteriorated notably. These changes can be attributed to the profound impact of the COVID-19 epidemic. Future research should focus on the long-term effects and lag impacts of the pandemic on physical fitness, particularly as the lasting consequences of COVID-19 continue to emerge. An urgent imperative exists to mitigate the documented adverse effects of the pandemic on university students’ physical fitness. Implementing multi-faceted interventions to enhance population-wide fitness levels, while establishing contingency protocols for future global health crises, represents a critical public health priority.

Supplemental Information

Supplemental Information 1 STROBE checklist.

Supplemental Information 2 Dataset.

The authors wish to thank the student participants for their contribution to the research, as well as the teacher investigators and researchers for all their efforts.

Additional Information and Declarations

Competing Interests

The authors declare that they have no competing interests.

Author Contributions

Jianzhong Sun conceived and designed the experiments, performed the experiments, prepared figures and/or tables, authored or reviewed drafts of the article, and approved the final draft.

Bin Qiao performed the experiments, analyzed the data, prepared figures and/or tables, and approved the final draft.

Chan Lin conceived and designed the experiments, analyzed the data, authored or reviewed drafts of the article, and approved the final draft.

Human Ethics

The following information was supplied relating to ethical approvals (i.e., approving body and any reference numbers):

This work was granted approval by the Human Experimental Ethics Committee of Chizhou University, with the ethics approval number CZ2018YJRC01.

Data Availability

The following information was supplied regarding data availability:

The data supporting the findings of this article are available in the Supplemental File.

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
