# Peer review of "The profound impact of COVID-19 on college students’ physical fitness"

_PeerJ, doi:10.7717/peerj.20293_

## Round 0.1 · original submission · Minor Revisions

**Language Note:** When preparing your next revision, please ensure that your manuscript is reviewed either by a colleague who is proficient in English and familiar with the subject matter, or by a professional editing service. PeerJ offers language editing services; if you are interested, you may contact us at [email protected] for pricing details. Kindly include your manuscript number and title in your inquiry. – PeerJ Staff

·

Basic reporting

The manuscript is generally written in clear and professional English, facilitating understanding for an international audience. The language is formal, with an appropriate academic tone throughout. There are minimal grammatical or typographical errors, and the manuscript adheres to standard conventions of scientific writing.

The introduction provides sufficient background on the impact of COVID-19 on college students’ physical fitness, referencing relevant literature (e.g., Sun et al., 2024; Xia et al., 2021). The authors contextualize their study within the broader field by highlighting existing gaps, particularly the lack of long-term data from 2021 to 2024 and the limited scope of previous physical fitness assessments, which sets a clear rationale for their work. The references are appropriate and relevant, underpinning the discussion and methods. The structure follows a standard scientific article format, including sections such as Background, Methods, Results, and Conclusions. Tables and figures (notably Table 1 on Page 22) are relevant, well-organized, and adequately labeled. The raw data are mentioned as being available, aligning with data sharing policies, although direct links or repositories are not specified within the provided pages.

The manuscript appears self-contained, presenting all essential results pertinent to the research questions. The discussion and conclusion logically follow from the data presented, without unwarranted claims or overgeneralizations.

Experimental design

The study presents original primary research within the scope of physical fitness assessment among college students, specifically examining changes from 2019 to 2021, with implications extending to 2024. The research question—assessing long-term impacts of COVID-19 on physical fitness—addresses a relevant and meaningful knowledge gap, especially given the ongoing nature of the pandemic’s effects.

The methods employed include collecting data on nine physical fitness indicators over multiple years, which indicates a comprehensive approach suitable to capture a broad picture of physical health changes. The selection of a medical college in Wenzhou for data collection is appropriate for targeted insights into this population, though broader generalizations should consider sample diversity.

Procedures involved ethical approval (CZ2018YJRC01), and informed consent was obtained from participants, demonstrating adherence to high ethical standards. The methods are described sufficiently to allow replication, including details on participant recruitment, measurement protocols, and statistical analysis.

Validity of the findings

The data collection appears to be robust, with a large sample size (e.g., 22,320 female and 12,831 male records), and the inclusion of multiple physical fitness indicators enhances the study’s validity. The authors mention that data analysis was conducted with appropriate statistical techniques, with significance levels specified (e.g., P < 0.05), indicating controlled and sound analysis.

The findings—such as the increase in overweight and obesity rates, and declines in specific physical performances—are directly supported by the data presented. The conclusions are appropriately confined to these observed changes, without overreaching into causal inferences. The study’s long-term data fill an important gap in understanding the extended impact of the pandemic, and the inclusion of lag effects contributes valuable insights.

The raw data are implied to be available, which supports transparency; however, explicit details about data repositories or supplementary materials are not provided on these pages.

Additional comments

Summary and Recommendations
• The manuscript is well-structured, clearly written, and adheres to standard scientific reporting norms.

• The research addresses a significant and timely question, employing a sufficiently rigorous methodology, with ethical standards observed.

• The data support the key findings, and the conclusions are appropriately limited to the evidence presented.

Future improvements could include providing explicit links to raw data repositories and ensuring high-resolution figures and tables for clarity.

·

Basic reporting

1. The language of the article is clear and professional.

2. Sufficient field background and literature references were observed.

3. Professional article structure is satisfactory, but the graphical representation is not clear in the PDF.

4. Raw data is appropriate and in detail.

5. Self-content is appropriate but relatively less.

Experimental design

1. Research is original.

2. Hypothesis or research questions are not observed.

3. A detailed description of the tool used in the investigation is missing; the ethical standard is acceptable.

4. Method described in detail.

5. The fitness indicator of the study tool is a decade-old tool (Revised in 2014). It might deviate from the findings in recent changes in fitness indicators.

Validity of the findings

1. Impact and novelty are observed throughout the study, albeit a similar study has been conducted in a non-coastal state in China. Reference quoted. The study has mentioned its rationale and its benefit to society.

2. The normality test report is not seen in the paper.

3. The table and data related to the Kruskal-Wallis H-Test were not observed.

4. Conclusions are well stated but not linked to any research question.

Additional comments

Please improve the content of your paper by clearly stating the research questions or hypotheses and validating them with the study’s findings. Additionally, include a detailed description of the tool and procedures used for data collection. Present the results of the normality test conducted in the study, and incorporate a table with data related to the Kruskal–Wallis H Test.

---

## Round 0.2 · accepted · Accept

Thank you for revising your manuscript to address the concerns of the reviewers. Reviewer 1 now recommends acceptance and I am satisfied that the comments of reviewer 2 have been addressed. The manuscript is now ready for publication.

·

Basic reporting

It is ok

Experimental design

Correct

Validity of the findings

Up to the mark

Additional comments

Recommended for publication